# Effectiveness of Mindfulness-Based Relapse Prevention in Individuals with Substance Use Disorders: A Systematic Review

**DOI:** 10.3390/bs11100133

**Published:** 2021-09-29

**Authors:** Eduardo Ramadas, Margarida Pedroso de Lima, Tânia Caetano, Jessica Lopes, Maria dos Anjos Dixe

**Affiliations:** 1Faculty of Psychology and Educational Sciences, University of Coimbra, 3000-115 Coimbra, Portugal; mplima@fpce.uc.pt (M.P.d.L.); taniasdcaetano@gmail.com (T.C.); 2Center for Innovative Care and Health Technology (ciTechcare), Polytechnic of Leiria, 2410-541 Leiria, Portugal; maria.dixe@ipleiria.pt; 3VillaRamadas International Treatment Center, Research and Innovation Department, 2400-121 Leiria, Portugal; jesslopes.uc@gmail.com

**Keywords:** mindfulness, MBRP, addiction, substance use disorder, systematic review, relapse prevention

## Abstract

Objectives: This systematic review aimed to understand the current state of the art about the effectiveness of mindfulness-based relapse prevention (MBRP) on individuals with substance use disorders (SUD), taking into account not only SUD variables (e.g., cravings, frequency of use) but also other relevant clinical variables (e.g., anxiety and depressive symptoms, quality of life). Methods: A comprehensive search of electronic databases was conducted to identify studies that investigate MBRP interventions on individuals with SUD. Studies that met the inclusion criteria were synthesized and assessed using systematic review methods. Results: Thirteen studies were included in the present review. The methodological quality of the included studies was moderately strong. Nine studies (69.2%) used the traditional 16 h MBRP program. Six studies (46.1%) chose to use a co-intervention treatment ranging from the treatment as usual (TAU) to cognitive behavioral therapy. All but one study indicated that their interventions produced positive effects on at least one addiction outcome measure. None of the interventions were evaluated across different settings or populations. Conclusions: Despite some heterogeneity regarding the type of MBRP program used, results support the effectiveness of these interventions in the SUD population, especially in reducing cravings, decreasing the frequency of use, and improving depressive symptoms.

## 1. Introduction

Substance use disorders (SUD) affect the lives of more than 35 million people worldwide according to 2019 data from the United Nations Office on Drugs and Crime [1]. A chronic disease, substance addiction is associated with significant costs not only for the individual but for society at large. It often starts with occasional substance use in a recreational context, followed by a progressive increase in frequency of use as the desired effects are diminished by habituation [2]. This process is mediated by genetic, developmental, and psychosocial factors that impact an individual’s relative vulnerability to the development of an addiction [3]. Although there are many different types of SUD, depending on the main substance of choice—with DSM-V offering diagnoses for the abuse of alcohol, cannabis, hallucinogens, opioids, and even caffeine, among others [4]—there are underlying mechanisms in addiction that appear to be common to all substances [3].

Given the prevalence and manifest individual and socioeconomic consequences of substance addiction, there has been a considerable effort by the scientific community to create and study effective treatments for SUD [5,6]. Despite this, relapse rates remain incredibly high, with data suggesting that more than two thirds of individuals return to substance use within a year of treatment [7]. The relapsing nature of substance addiction has made it clear that relapse prevention is an essential treatment component for long-term recovery, shifting attention to interventions that propose to address this need.

The most common intervention targeting relapse prevention is based on the identification of idiosyncratic high-risk situations that can increase the probability of relapse and the development of specific strategies that the individual can resort to when exposed to them [8]. However, recently, other types of interventions aiming to improve the effectiveness of SUD treatment and address relapse prevention have been emerging, as is the case with mindfulness-based interventions [9,10].

Mindfulness is a mental state that involves an awareness of the present moment and a non-judgmental posture towards internal and external phenomena. Practicing mindfulness has been shown to help regulate attention, cultivate acceptance, and develop the ability to observe inner experiences (thoughts, feelings, and physical sensations) without judgment [11,12]. In the context of addiction treatment, it can promote awareness of external and internal triggers for addictive behaviors and improve tolerance to uncomfortable emotional, cognitive, and physical experiences [13]. It can also have a significant effect on decreasing cravings after treatment [14].

Mindfulness-based relapse prevention (MBRP) is an intervention that integrates mindfulness meditation with traditional relapse prevention techniques. It has three main components: formal mindfulness practice, informal practice, and coping strategies. The formal practice includes body scan, sitting meditation (breath, body, thoughts, and emotions), mountain meditation, and loving-kindness meditation. The informal practice includes urge surfing, mindfulness of daily activities, SOBER breathing space, and mindful movement [15,16,17]. Some of the informal practices can be used as coping strategies due to the fact that they can help the patients cope with difficulties, emotional challenges, or stress [18]. As such, this last component intends to adapt informal practice into practical strategies the individuals can use in real-life situations.

The original MBRP program has a total of 16 h divided into 8 weekly sessions (2 h each) but can be adapted according to the target population and the researchers’ goals [19,20]. The sessions include mindfulness practices and exercises that aim to increase the individual’s awareness of internal (emotional and cognitive) and external (situational) cues related to substance use and help in the development of appropriate coping strategies [21].

The first three sessions of the program focus on raising the individual’s awareness of environmental triggers and reactions that may occur in response to them. The first session centers on the individual’s habits, which occur in autopilot mode; the second centers on increasing the awareness of triggers and cravings; and the third centers on the promotion of mindfulness practice in daily life. Sessions four, five, and six focus on mindfulness in the context of relapse prevention. The fourth session centers on the use of mindfulness in high-risk situations; the fifth session intends to teach acceptance and skillful action; and the sixth session centers on the concept of defusion, that is, the ability to see thoughts only as thoughts and not as reflections of reality. Finally, the last two sessions focus on social and environmental factors and have the goal of guiding the application of what was learned to everyday life [15,22].

The aim of this review was to evaluate the effectiveness of MBRP in individuals with SUD. Although the search conducted on the Web of Science and Cochrane Database of Systematic Reviews revealed the existence of a systematic review/meta-analysis looking into this same question [23], we considered that the interest this topic has gathered in recent years justified another review. It is not only likely but probable that in the four years that have passed since Grant et al. [23] conducted their review, new relevant studies have been added to the literature. Given the rapid growth of knowledge concerning this type of intervention, we also hope that more recent studies have taken into account some of the limitations found in their earlier counterparts.

To better understand the effectiveness of MBRP in this population, we considered the following questions:IDo MBRP interventions lead to significant improvements in SUD variables (e.g., cravings, frequency of use) in individuals with SUD?IIDo MBRP interventions lead to significant improvements in other relevant clinical variables (e.g., anxiety, depression, quality of life) in individuals with SUD?IIIHow do MBRP interventions compare to other evidence-based interventions for individuals with SUD?

## 2. Materials and Methods

### 2.1. Research Strategy

This systematic review was not pre-registered, and the results should be considered exploratory.

Through the research performed on the Cochrane Central Register of Controlled Trials (CENTRAL), JBI Database of Systematic Reviews and Implementation Reports, Database of Promoting Health Effectiveness Reviews (DoPHER), Evidence for Policy and Practice Information and Coordinating Center (EPPI-Center), and PROSPERO International Prospective Register of Systematic Reviews, no systematic literature review was found after the year 2017.

Initially, a general search was carried out in Cochrane, Pubmed, and Web of Science, after which the terms used in the title and abstract of the articles were examined. In a second phase, using the identified keywords and descriptors, research was conducted on all databases included in this review (Table A1). Due to the multiplicity of concepts to describe the term (addiction) we used a global MeSH term in our research. In the last phase of the research, we used the bibliography of the selected studies to search for additional/complementary ones.

### 2.2. Study Screening and Selection

The inclusion criteria were developed based on the PICO system used to form the research question.

Inclusion criteria were studies that included participants with minimum age of 18 years (female or male) with diagnosis of substance use disorder (alcohol, opioids, cannabis, and/or stimulants). Regarding the intervention, studies that used the MBRP as an individual and/or complementary intervention tool were included in the present review. As for the comparator, all types of interventions were accepted, namely the TAU. As for outcomes of interest, we intend to investigate if the MBRP interventions are effective in this type of population based on indicators supported by other studies in the field [9,23].

We distinguished the following categories of study designs to be included in the present review: experimental designs (randomized controlled trials [RCT]) and quasi-experimental studies. Studies written in English, Spanish, and Portuguese were included.

Exclusion criteria were omission of diagnosis of substance use disorder, intervention without recourse to the MBRP, inclusion of a population under 18, study design other than experimental or quasi-experimental, and studies written in a language other than Portuguese, English, or Spanish. The differences found between reviewers during this process were resolved through dialogue and, when necessary, through a third reviewer.

### 2.3. Data Extraction and Quality Assessment

Data were extracted by two independent reviewers, with emphasis on specific details related to the specificity of the population, the type of co-intervention used, the type of comparator, the study design, and relevant results underlining the objective and conducting question of the review.

In order to minimize the risk of bias among the selected studies, the Joanna Briggs Institute Checklist for Randomized Controlled Trials was used for the critical analysis of the RCT included. For the analysis of the quasi-experimental studies, the Checklist for Quasi-Experimental Studies was used. To examine the methodological quality of the included studies, the “Study Appraisal Checklist” was used. This checklist is composed of 27 items divided into 5 categories (reporting, external validity, bias, confounding, and power) with a higher score being equivalent to a higher methodological quality. According to the checklist development study [24] and the other systematic review [25] using this same instrument, the cut-off point of 14 was used as a guideline to define low/good methodological quality of the studies.

The differences found between reviewers during this process were resolved through dialogue and, when necessary, through a third reviewer. The risks of bias for each type of study included were examined in detail (e.g., selection bias). An individual assessment of the risk of bias was performed for each of the included studies. These results are reported in the Methodological Quality section.

### 2.4. Data Analysis

The influence of the interventions carried out using the MBRP and its effects on the population was described in a narrative format for each of the included interventions. A description was made taking into account the year in which the studies were carried out.

The data analysis was described based on the following topics: aim of the study, type of MBRP program used, the researcher administrator, type of co-intervention and comparator used, the existence of follow-up, and the respective results. Regarding the results, they were presented taking into account the variables of interest, namely the relapse rate, frequency of use, cravings/desire to use, and depression and anxiety symptoms.

## 3. Results

### 3.1. Search

A total of 215 articles were identified as potentially relevant to the present review (Figure 1). Of these, 42 were excluded due to being duplicate studies. The titles and abstracts of the remaining articles were examined taking into account the inclusion and exclusion criteria, and 143 more studies were excluded. From the 30 studies selected for the full text analysis, 17 were excluded for not meeting all the inclusion criteria (seven did not mention a diagnosis of substance use disorder, six did not have an experimental study design, three did not present the outcomes found, and one had a different population). Of the 13 studies included, 10 (76.9%) are RCT and 3 (23.0%) are quasi-experimental studies without a control group.

### 3.2. Characteristics of Included Studies

All the studies included were written in English and, as previously mentioned, published between 2016 and 2020. We decided to include studies since 2016, as some studies published this year may not have been considered in the previous review. Regarding the country of origin, seven studies (53.8.5%) were carried out in the United States [19,20,26,27,28,29,30], four (30.7%) in Iran [31,32,33,34], and two (15.3%) in France [22,35]. The sample sizes ranged from 6 to 191. The type of population used was different among the studies: four (30.7%) used only individuals with alcohol use disorder [19,20,22,29], three (23.0%) included individuals with substance use disorder related to methadone [31,32,33], and two (15.3%) used individuals with a stimulant and opioid use disorder, respectively [26,30]. On the other hand, three studies (23.0%) had heterogeneous samples, including individuals with various substance use disorders [28,34,35]. One (7.6%) of the studies did not provide information on the characteristics of the sample used [27]. Table A2 presents the synthesized information of the included studies.

### 3.3. Methodological Quality

The design of the included studies consisted of 10 (76.9%) RCT [19,20,26,27,28,29,31,32,33,34] and 3 (23.0%) quasi-experimental studies [22,30,35]. The methodological quality of the studies according to the Study Appraisal Checklist varied between 10 and 25 (maximum = 28) with an average of 17.3 [SD = 3.9]. The studies with the lowest and highest quality had values of 37.0% and 92.5%, respectively. Only two studies were categorized as being of low methodological quality [22,35]. A decision was made to include these studies with the goal of considering a greater variety of study designs. The specific methodological quality flaws that were found in the great majority of the included studies are related to the distribution of the main confounders in each group (*n* = 8), the adverse events (*n* = 7), the representativeness of the sample included (*n* = 12), the lack of blindness of the assessors (*n* = 10), and no calculation of the sample power (*n* = 9). Table A3 displays the score of each study.

With regard to the methodological quality of the RCT included, the questions (Q2, Q4, Q5, and Q6) of the used checklist required refinement in the majority of the studies, only being clear in three of them (Table A4). These questions clarify, respectively, whether the allocation to the treatment group was concealed (Q2), whether the participants were blind to the assigned condition (Q4), whether the researchers who applied the intervention were blind to the treatment assignment (Q5), and whether the outcome assessors were blind to the treatment assignment (Q6).

In the study conducted by Glasner et al. (2017; [26]), the allocation into the respective groups was concealed from participants but not from the practitioners. On the other hand, in the study by Witkiewitz et al. (2019; [19]), although both participants and practitioners were blinded to the assigned condition, the researchers responsible for analyzing the outcomes were not. Regarding the reliability of the outcomes and the statistics used, the studies by Yaghubi et al. (2017; [32]) and Zgierska et al. (2017; [20]) were not clear about either the statistical analyses used or the outcome measures considered. The study by Zemestani and Ottaviani (2016; [34]) presented greater objectivity and clarity in all the procedures performed.

Regarding the methodological quality of the quasi-experimental studies included, seven of the nine questions included in the checklist used were clear in all studies (Table A5). These questions referred to the clarity of the study with regard to cause and effect (Q1), whether the participants were included in any type of comparison (Q2), if there was a control group (Q4), if there were multiple evaluations of results before and after the intervention (Q5), if follow-up was performed (Q6), whether the results of the included participants were evaluated in the same way (Q7), and whether appropriate statistical tests/procedures were used (Q9). In the study of Von Hammerstein et al. (2019; [22]), it is not clear if the participants received other intervention in combination with the intervention of interest. In the studies by Zullig et al. (2018; [30]) and Biseul et al. (2017; [35]), it is not clear how the reliability of the results measured was guaranteed. Neither study presented a control group.

### 3.4. Intervention Characteristics

Despite the heterogeneity of the included studies, it was found that nine used the traditional 16 h MBRP program [22,27,28,30,31,32,33,34,35]. The remaining four chose to use a version of the program adapted to the population and/or study objective [19,20,26,29]. There was also some diversity between studies regarding the length of the intervention program and the length of the sessions. While the majority of the studies (84.6%) used the standard 16 h intervention, with a duration of 120 min for each session [20,22,27,28,29,30,31,32,33,34,35], two studies chose to adapt the MBRP program according to the desired intervention characteristics and/or time available for it. One study (7.6%; [26]) used the 10 h MBRP program, with each session having a duration of 75 min. Another study (7.6%; [19]) chose to use a 12 h program, with each session having a duration of 90 min.

At the intervention level, most studies chose to use a co-intervention treatment. Three studies (23.1%) used medication-assisted treatment (e.g., methadone-therapy) as co-intervention [30,31,32], with two of those also using it as a comparator [30,31]; two (15.3%) used TAU as both co-intervention and comparator [20,29]; one (7.6%) used transcranial direct current stimulation as a co-intervention and a sham version of the same intervention as a comparator [19]; one (7.6%) study chose a contingency management intervention that was also a comparator together with a health education intervention [26]; and one study (7.6%) chose to use cognitive behavior therapy as a co-intervention and did not use a comparator [35].

The remaining studies did not use any type of co-intervention, with one (7.6%) using only TAU as a comparator [34], one (7.6%) using a traditional relapse prevention intervention [27], and another (7.6%) using both TAU and relapse prevention [28] as comparator interventions. Two studies (15.3%) chose not to use either a co-intervention or a comparator [22,33].

All but one (7.6%) study [30] used follow-up assessments in order to analyze the consistency and reliability of the outcomes obtained.

Table A6 presents a summary of the results described.

### 3.5. Intervention Outcomes

Of the 13 included studies, only one did not report significant improvements in at least one of the outcome measure [29]. The studies by Witkiewitz et al. (2019; [19]), Roos et al. (2020; [28]), and Greenfield et al. (2018; [27]) presented a significant decrease in the substance use frequency of the participants in the MBRP group: (SE = −0.535 (0.16), *p* = 0.001), (IRR = 0.000003, CI (−0.032, 0.021), and (IRR = 0, 95% CI: 0,0). Another three studies [31,33,34] showed a significant reduction in the desire to use following the MBRP intervention: (F (2,144) = 35.90, *p* < 0.0001) and (F = 374.22, *p* = 0.00). Moreover, Zemestani and Ottaviani (2016; [34]) also reported a beneficial impact of MBRP on anxiety (F (2,144) = 43.96, *p* < 0.0001) and depressive (F (2,144) = 30.73, *p* < 0.0001) symptoms. Two other studies found similar results regarding the significant improvement of depressive complaints [22,35].

Lastly, two studies presented significant and positive changes on impulsivity levels [32] and quality of life [33].

## 4. Discussion

The objective of the present review was to evaluate the effectiveness of MBRP on individuals with SUD. A majority of the included studies [19,22,26,27,28,30,31,32,33,34,35] found significant improvements in at least one SUD or clinical variable following an MBRP intervention. This was true not only for studies that used the original MBRP program but also for those that chose a modified version (e.g., different program length or session duration).

Regarding the impact of MBRP on SUD variables, the results indicate that participants that received the intervention had a significant decrease in frequency of use, both measured by days of heavy drinking [19,22,27,28] and number of drinks per day [19] for alcohol and number of using days/substance misuse for other substances [27,35]. There was also a significant decrease in cravings/desire to use [31,33,34], withdrawal symptoms [31], and probability of relapse [32].

Concerning other clinical variables, the results suggest that participants who received an MBRP intervention saw a significant decrease of anxiety [22,34] and depression [22,26,30,34,35] symptoms as well as reduced impulsivity levels [32]. Moreover, they showed a significant improvement in quality of life [33] and coping capacity over time [28]. These findings seem to be congruent with the subjective experience of the participants which, as found by Zgierska et al. (2017; [20]), presented an overall satisfaction with the intervention and believed it to be helpful for their disorder.

Only three studies [27,28,34] chose to apply the MBRP program without a co-intervention while providing a comparator group. Two other studies used TAU as a co-intervention and comparator [20,29]. This made it difficult to properly compare the effectiveness of MBRP against that of other evidence-based treatments. Still, the results are relevant to understand if new knowledge on the relative effectiveness of MBRP has been created since the last systematic review conducted by Grant et al. [23].

Supporting the conclusions from the previous review [23], Zgierska et al. [29] found that MBRP plus TAU was not more effective than TAU in individuals with alcohol dependence. Even though there were no significant group differences, it should be noted that individuals from both groups showed favorable outcomes. On the other hand, Roos et al. [28] and Zemestani and Ottaviani [34], who also compared MBRP with TAU, found different results. Roos et al. [28] reported an interaction by affective symptoms that predicted increases in approach coping, which in turn led to more favorable substance use related outcomes (i.e., fewer heavy drinking days and substance use problems at 12 months). The mediator effect of increased approach coping was only found for individuals with high base-line affective symptoms (as opposed to low or moderate). Similarly, Zemestani and Ottaviani [34] found that MBRP was significantly more effective than TAU in reducing rates of depression, anxiety, and cravings in individuals with SUD and comorbid depression. Finally, Greenfield et al. [27] compared the current most used form of relapse prevention [8] with MBRP and found significant differences regarding the decrease of frequency of use.

The results from the studies by Roos et al. [28] and Zemestani and Ottaviani [34] are particularly interesting. They indicate that MBRP may be especially useful for individuals who present SUD in comorbidity with severe psychiatric symptomatology, namely those with affective disorders. This could be explained by the impact mindfulness has been shown to have in facilitating emotional regulation. Hölzel et al. [36], for example, have proposed that the effects of mindfulness are a result of changes in mental and brain functions pertaining to, among other self-regulatory systems, attention and emotional regulation. Similarly, Vago and Silbersweig [37] suggested that changes in self-regulation and self-awareness resulting from mindfulness reflect modulation in neurocognitive networks associated with various dimensions, including emotional regulation. Some authors (e.g., [38,39]) have even argued that mindfulness might employ a unique strategy of emotional regulation based on bottom-up systems (contrary to cognitive reappraisal, which relies on top-down systems). Bottom-up pathways are thought to be responsible for emotional processing (generation of arousal and affect; [40]) and have been linked to some limbic regions and the striatum [41].

Interestingly, bottom-up along with top-down processes have also been hypothesized as possible mediators to the effects of mindfulness practice on SUD [40]. This is at least in part explained by the role these and associated pathways are proposed to play on the reactivity to drug stimuli, drug cravings, and relapse. Despite their believed relevance, so far these pathways have barely been researched.

As with all studies, the present review has limitations that need to be considered when interpreting its results. It is important to highlight the presence of methodological weaknesses in some of the included studies, such as a reduced sample size in those using a quasi-experimental design [22,30,35]. Given that these weaknesses can skew the results obtained, it is not possible generalize them and conclude the effectiveness of the intervention under study. Another aspect to consider is the significant heterogeneity found between studies in relation to the main objectives, target population, and type of intervention, which invalidated the option of conducting a meta-analysis. Beyond the characteristics of the studies included, another limitation of this systematic review was the reduced number of databases used for the article search. A broader search might have led to the discovery and inclusion of more studies and, as result, a more complete account of the current state of the art about this subject matter.

In conclusion, we believe the results of the present review indicate that MBRP has a significant positive impact on various substance use and clinical variables. Moreover, even though we could not properly evaluate the comparative effectiveness of MBRP in relation to other evidence-based treatments, the results suggest that this intervention may be of particular value to the treatment of individuals with SUD and comorbid psychiatric symptomatology. Future studies should not only continue to investigate the effectiveness of MBRP in individuals with SUD (with and without psychiatric comorbidity) but also explore possible mediating variables that may allow for a better understanding of it. Furthermore, we believe that future research into the effectiveness of MBRP would greatly benefit from considering neural measures in order to deepen the knowledge of the neural circuit mechanisms involved in the practice of mindfulness and how these relate to relapse prevention. There were few studies that considered neural responses in their results, with only one study of interest in this area [19].

## Figures and Tables

**Figure 1 behavsci-11-00133-f001:**
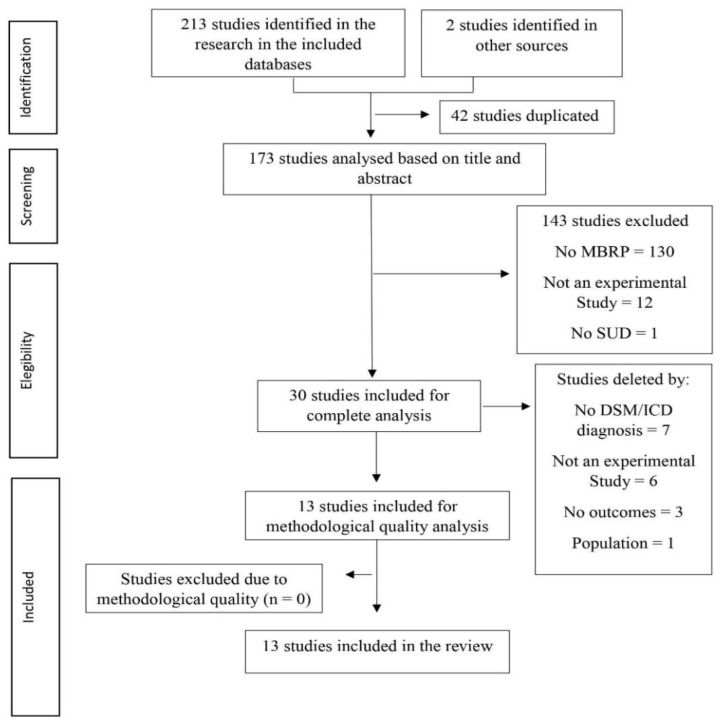
PRISMA Flow Diagram.

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
