# Peer review of "Effectiveness of Mindfulness-Based Relapse Prevention in Individuals with Substance Use Disorders: A Systematic Review"

_behavsci, 2021, doi:10.3390/bs11100133_

Round 1
Reviewer 1 Report
The authors have done a great deal of work and that is very welcomed. I read it with great interest, I hope that the readers will also be reading it with interest. Mindfulness practice is increasingly used to reduce various disorders.
However, I do have some comments on this article.
The authors referred to Sinha, R. [7], referring to relapse, but the author herself has referred to other authors, and this article discussed biological markers, so it would be better to refer to the original sources.
The authors mention that mindfulness includes three main components, but described only two. If 3 components were mentioned, coping strategies should also be described briefly.
Results: It is not clear why the authors split the tables (6, 7, 8) rather than merged them into one.
Discussion: How do the authors explain why their results differ from the previous systematic review (Grant et al., 2017)? Sentence [308] does not correspond to what has been said before/above, because most articles did not mention co-intervention.
The bibliography should be arranged according to a unified system.
Author Response
We appreciate your feedback and hope that the changes we made sufficiently address your concerns and suggestions.
Point 1: The authors referred to Sinha, R. [7], referring to relapse, but the author herself has referred to other authors, and this article discussed biological markers, so it would be better to refer to the original sources.
Response 1: We eliminated the Sinha, R. [7] reference and added the reference of the original source.
Point 2: The authors mention that mindfulness includes three main components, but described only two. If 3 components were mentioned, coping strategies should also be described briefly.
Response 2: We had indeed forgotten to describe one of the three main components of MBRP. We added two sentences to this paragraph to provide a brief description of coping strategies in the context of MBRP.
Point 3: Results: It is not clear why the authors split the tables (6, 7, 8) rather than merged them into one.
Response 3: We had divided the tables (6,7,8) because we thought it might make it easier to read the results. We have merged them into one and updated the annexes.
Point 4: Discussion: How do the authors explain why their results differ from the previous systematic review (Grant et al., 2017)? Sentence [308] does not correspond to what has been said before/above, because most articles did not mention co-intervention.
Response 4: With regards to the discussion, we made quite a few improvements. The sentence you referred to, which mentioned the fact that most studies had chosen to use a co-intervention, was not clear. We ended up not only changing that sentence but also improving the description on the result’s section that pertained to these variables (co-intervention and comparator). We also extended the paragraph in the discussion to better explore the results pertaining to the effectiveness of MBRP in comparison with other interventions. We believe this allowed for a more nuanced comparison between our results and the conclusions of the previous systematic review conducted by Grant et al. (2017).
Point 5: The bibliography should be arranged according to a unified system.
Response 5: We have reviewed the bibliography and corrected any mistakes so that it is arranged according to a unified system.
Reviewer 2 Report
The authors reviewed the effectiveness of mindfulness in the prevention of drug relapse. Overall, it is a nice review that has summarized many important points. However, my only comment is the lack of discussion on potential neural circuit mechanisms involved in the process of mindfulness.
It would be interesting to talk about the potential circuit involved. The most obvious one would be mesolimbic DA system as well as the RMTg which directly inhibits VTA DA neuorns.
Author Response
We appreciate your feedback and hope that the changes we made sufficiently address your concerns and suggestions.
Point 1: However, my only comment is the lack of discussion on potential neural circuit mechanisms involved in the process of mindfulness.
It would be interesting to talk about the potential circuit involved. The most obvious one would be mesolimbic DA system as well as the RMTg which directly inhibits VTA DA neuorns.
Response 1:
Since the studies we included in the review (with one exception) did not consider neural measures, we had not thought about discussing potential neural circuit mechanisms involved in mindfulness practice. However, we do think this addition strengthened the discussion.
We added two paragraphs exploring the neural pathways thought to be affected by mindfulness practice and touched upon their connection to substance abuse and relapse.
Please let us know if you think we should deepen this part of the discussion or if you believe our additions are sufficient.
Point 2 (checklist): English language and style are fine/minor spell check required
Response 2: We also reviewed the text and tried to improve on the English.
Round 2
Reviewer 1 Report
Thank authors for making corrections!
I think this manuscript will be interesting for specialists.
Reviewer 2 Report
I have no further comments.